# Probiotics in the Prevention and Treatment of Gestational Diabetes Mellitus (GDM): A Review

**DOI:** 10.3390/nu14204303

**Published:** 2022-10-14

**Authors:** Klaudia Kamińska, Dominika Stenclik, Wiktoria Błażejewska, Paweł Bogdański, Małgorzata Moszak

**Affiliations:** 1Student Scientific Club of Clinical Dietetics, Department of the Treatment of Obesity and Metabolic Disorders, and of Clinical Dietetics, Poznań University of Medical Sciences, Szamarzewskiego 82/84, 60-569 Poznan, Poland; 2Department of Treatment of Obesity, Metabolic Disorders and Clinical Dietetics, Poznan University of Medical Sciences, 60-569 Poznan, Poland

**Keywords:** gestational diabetes, gut microbiota, probiotics

## Abstract

Gestational diabetes mellitus (GDM)is one of the most common perinatal pathologies, with a prevalence of 5–20% depending on the population or diagnostic standards. It is diagnosed when glucose intolerance is first detected during pregnancy. In the pathogenesis of GDM, genetic, environmental, and pregnancy-related factors (excessive fat storage and increased adipokine and cytokine secretion) play important roles. A growing amount of scientific data has indicated the role of gut microbiota (GM) dysbiosis in the development of glucose intolerance during pregnancy. Previous studies have indicated that, in comparison to healthy pregnant women, GDM individuals have a greater abundance of bacteria belonging to the genera *Ruminococcus, Eubacterium*, and *Prevotella* and a lower number of bacteria belonging to the genera *Bacteroides, Parabacteroides, Roseburia, Dialister*, and *Akkermansia*. Recently, many studies have focused on treating GDM with methods targeting GM. Several previous studies have analyzed the effect of probiotics on the course of GDM, but their data are inconclusive. In view of this state, the aim of the study was to collect and comprehensively discuss current knowledge regarding the role of probiotic supplementation in preventing and treating GDM. According to the analyzed data, probiotics have a positive influence on glycemic control and are a promising tool for lowering the frequency of GDM. However, further studies must be conducted to determine the optimal model of probiotic therapy (strain, dose, time of intervention, etc.) in pregnant women with GDM.

## 1. Introduction

The necessity of providing an ideal environment and optimum conditions for a developing fetus induces many anatomical and metabolic changes in maternal physiology. In the first few weeks of pregnancy, significant changes can be observed in the circulatory system, the respiratory system, and kidney function. To meet the growing cellular demand for oxygen and nutrients, blood volume increases, which entails an increase in heart output (by around 40%), heart rate, peripheral vasodilatation, stroke volume (by up to 30%), hydrostatic capillary pressure, and renal and uterine flow. As the increased red blood count is not as intense as the changes in the circulating blood volume, many pregnant women develop anemia. The pregnancy period is also characterized by a progressive decrease in platelet count and a hypercoagulable physiological state [1], pregnancy-influenced hyperventilation [2], decreased functional residual capacity, and modulated inspiratory reserve volume. In physiological pregnancy, an altered hormonal state (involving human chorionic gonadotropin (hCG), estrogen, progesterone, and thyroid-stimulating hormone (TSH)) leads to gastrointestinal reactions, such as vomiting and nausea. The increased concentration of adrenocorticotropic hormone (ACTH) and cortisol leads to hypercortisolism. Pregnancy also affects many metabolic pathways. For example, pregnancy promotes hypertriglyceridemia and hypercholesterolemia, decreases protein catabolism, and modulates glucose metabolism. In normal pregnancy, mild fasting hypoglycemia and prolonged postprandial hyperglycemia help optimize the flow of glucose to the developing fetus. From the second trimester, with peaks in the third trimester, there is a gradual development of insulin resistance (IR), resulting from increased secretion of diabetogenic hormones (progesterone, estrogens, prolactin, and cortisol), as well as hormones specific to pregnancy; i.e., human placental lactogen (HPL) and human placental growth hormone (hPGH). Previous studies have described how changes in hormone concentrations might impair insulin signaling; however, no single hormone has been identified as a pregnancy IR marker [3].

Up to a point, insulin sensitivity impairment is compensated for by an increase in insulin production by pancreatic beta cells, but in women with impaired glucose tolerance, this mechanism is insufficient to protect against the development of further disturbances in carbohydrate metabolism [4]. Environmental factors, such as maternal obesity, diet, and physical activity; pregnancy-related factors, such as excessive fat storage and increased secretion of adipokine and cytokines (e.g., tumor necrosis factor-α(TNF-α) and interleukin-6(IL-6)) [5]; and genetics also contribute to the development of IR in pregnancy [3]. A large amount of scientific data from the last two decades has also indicated the role of gut microbiota in the development of gestational glucose intolerance during pregnancy [6,7,8,9].

GDM is diagnosed when glucose intolerance is first detected during pregnancy [10]. GDM is one of the most frequent perinatal pathologies, with a prevalence of 5–20% depending on the population and diagnostic standards. The reasons for the growing incidence of GDM are the older age of women conceiving and the general trend toward an increasing number of overweight/obese women with metabolic disorders [11].

Hyperglycemia in pregnancy has serious and long-term clinical implications for the mother, the fetus, and the future of the newborn baby. The most common is macrosomia, which can lead to other perinatal complications, such as preeclampsia, elevated risk of Cesarean delivery, fetal death, birth injury (shoulder dystocia), and respiratory distress [12]. Previous studies have revealed that GMS leads to a predisposition to neonatal hypoglycemia, hyperbilirubinemia, and hypocalcemia [13]. It is worth emphasizing that, regardless of the severity of the course of perinatal hyperglycemia, maternal GDM is a significant risk factor in the event of excessive body mass, diabetes, hypertension, and dyslipidemia in childhood and adolescence [14,15,16]. Adverse maternal effects from GDM have also been reported, especially pregnancy-induced hypertension, preeclampsia, and eclampsia, as well as increased risks of developing breathing disorders, circulatory disorders, obesity, type 2 diabetes, and cardiovascular diseases in the future [17]. Due to health complications related to untreated GDM, all pregnant women should undergo screening tests at 24–28 weeks of gestation, and if GDM is diagnosed, the women should be promptly directed to a specialist center where they can be treated by professional health-care providers. A low-glycemic, carbohydrate-controlled diet and physical activity for a minimum of 150 min a week form the basis of the treatment of gestational hyperglycemia. Swimming, walking, yoga, and other types of low-/moderate-intensity aerobic exercise are optimal forms of exercise for pregnant women; however, the form and intensity of physical activity should be decided in consultation with the doctor. If treatment through lifestyle changes proves insufficient, pharmacological treatment based on insulin therapy should be implemented. Other important elements of GDM therapy are self-monitoring of blood glucose, intensive patient education, and management of gestational weight gain. Proper management of GDM ensures the optimal course of pregnancy and determines lifelong outcomes in GDM women and their children [18]. This is the reason researchers are looking for other safe methods (i.e., methods other than pharmacotherapy and lifestyle changes) that could prevent or treat GDM. Accessible scientific evidence from animal studies and clinical trials has confirmed the relationship between dysbiosis and the development of metabolic disorders, including GDM. Therefore, many existing studies have focused on therapies targeting GM [19,20].

The gut microbiota (GM) isa vast collection of microorganisms composed of bacteria, fungi, archaea, and viruses inhabiting the gastrointestinal tract. They form a complex ecosystem involved in, for example, harvesting energy, metabolizing nutrients and drugs, synthesizing vitamins, defending against inflammation, and protecting against pathogens [20,21]. The action of the GM is not limited to the gastrointestinal tract. Currently, the gut–brain, gut–liver, gut–skin, and gut–heart axes are the subjects of many studies [22]. We can distinguish two states of the GM. The state of emboss, in which the GM is in a quantitative and qualitative balance, is characterized by a predominance of beneficial species and ensures the maintenance of physiological reactions and homeostasis. The state in which such homeostasis is disturbed is called dysbiosis [23]. The GM varies in physiology from person to person depending on numerous factors, such as diet, lifestyle, previous pharmacotherapy, and age, thus constituting a unique and individual “fingerprint”. Pregnancy, both normal and complicated with diseases, also changes the composition and activity of the GM because of fat mass gain, hormonal changes, and increased release of pro-inflammatory cytokines [24]. Compared to the state before conception, in pregnancy, the GM is characterized by an increase in bacteria belonging to the phyla Proteobacteria and Actinobacteria, with a simultaneous depletion of the beneficial *Roseburia intestinalis* and *Faecalibacterium prausnitzii* [25].

Numerous studies have been conducted during the last two decades to compare the GMs of GDM and healthy pregnant women (Table 1). A depletion in the diversity of bacterial species, shifts in the abundance of specific bacterial taxa, and consequent changes in microbiota metabolic activity have been observed [26]. 

In a study by Cortez et al. [28], in comparison to healthy pregnant women, GDM patients had a higher abundance of *Ruminococcus, Eubacterium*, and *Prevotella* and a lower number of bacteria belonging to the genera *Bacteroides, Parabacteroides, Roseburia, Dialister*, and *Akkermansia*. *Ruminococcaceae* is involved in energy metabolism, insulin signaling, and inflammatory processes, and an increase in the relative abundance of *Ruminococcaceae* correlated with fasting glucose concentration and IR ledto a greater risk of GDM development [32]. A study involving 52 pregnant women showed that GMs in GDM patients with uncontrolled blood glucose were characterized by the enrichment of *Blautia* and *Eubacteriumhalliigroup* and depletion of *Faecali bacterium, Subdoligranulum*, *Phascolarctobacterium*, and *Roseburia.* Bacteria belonging to the genus *Faecalibacterium* were found to be important producers of short-chain fatty acids (SCFAs), especially butyrate, that promote *β*-cell differentiation and proliferation, improving insulin resistance [33]. In a study by Ye et al. [29], the number of *Feacali bacterium* was negatively correlated with the fasting blood glucose level, while the number of *Blautia* was positively correlated with the fasting blood glucose level. What is more, the study showed that changes in the GM may lead to GDM development by deregulating the peroxisome proliferator-activated receptor (PPAR) signaling pathway, the insulin signaling pathway, and the adipocytokine signaling pathway [29]. SCFA deficiency also contributes to the loss of tight connections and an increase in the permeability of enterocytes. In such a situation, the absorption of bacterial endotoxins, including lipopolysaccharide (LPS), increases, leading to the production of pro-inflammatory cytokines, an additional factor predisposing women to the development of IR and GDM [29]. A relationship between changes in the abundances of *Blautia, Butyricicoccus, Clostridium, Coprococcus, Dorea, Faecalibacterium, Ruminococcus,* and *Lachnospiraceae* (reduced numbers) and of *Collinsella* and *Rikenallaceae* (increased numbers) and GDM development has also been proved [27,30]. Importantly, scientific data show that the state of dysbiosis during GDM can be modified through dietary therapy. A prospective observational study by Ferrocino et al. [34] showed that a nutritional intervention based on the classic recommendations of a carbohydrate-controlled diet led to increased microbiota α-diversity (*p* < 0.001), elevated Firmicutes, and reduced numbers of Bacteroidetes and Actinobacteria. A higher ratio of Bacteroidetes to Firmicutes correlates with increased plasma glucose levels [35]. Additionally, Ferrocino et al. have observeda significant negative correlation between *Feacalibacterium* and fasting blood glucose; between *Collinsella* (directly) and *Blautia* (inversely),insulin, and HOMA-IR; and between *Sutterella* and C-reactive protein levels [34]. Disturbances in the composition of the intestinal microbiota occurring in the course of GDM may imply an increased risk of developing metabolic disorders in the future; therefore, probiotics, as a recognized modulator of GM, seem to be a potentially promising target in GDM [7].

In view of this context, the aim of this study was to collect and comprehensively discuss current knowledge regarding the role of probiotic supplementation in GDM prevention and treatment.

## 2. Materials and Methods

### 2.1. Eligibility Criteria

In this review, we included original studies on probiotic supplementation in GDM involving human subjects and published in English from 1 January 2010 to 30 July 2022 (a study from 2008 was added because it was related to a study from 2010).

The eligibility criteria were as follows: (1) the studies were randomized controlled trials (RCTs) comparing probiotics with placeboes, but we did not exclude interventions based on probiotics along with other supplements; (2) diagnostic criteria for GDM included International Association of Diabetes and Pregnancy Study Groups (IADPSG) criteria, Carpenter and Coustan criteria, and the criteria of the Fourth International Workshop–Conference on Gestational Diabetes Mellitus, but we did not exclude articles in which pregnant women were stated to be diagnosed with GDM but no diagnostic criteria were described; (3) the women were ≥ 18 years old; (4) the treatment duration was more than7 days; (5) the study group had over 10 participants. Only human studies were eligible; animal studies were excluded.

During the research, articles with data duplication, studies not meeting the inclusion criteria (such as case reports, studies where only the abstract was available, and non-English studies), and studies with inadequate information were removed.

### 2.2. Selection Process

Two reviewers (M.M. and K.K.) independently completed screening, data extraction, and quality assessment. In all, 96 records were retrieved. The “human species” criterion was added using automation tools, and 14 studies were removed. After the abstracts were read (W.B. and K.K.), another 21 studies were removed. The screening process was performed mechanically by checking the date of publication, GDM diagnosis, and the intervention protocol. The articles were browsed through and another48 were eliminated, including 1 study from 2008.The process left us with 12 studies that we included in our analysis.

### 2.3. Data Collection Process

This review was performed following the Preferred Reporting Items for Systematic Reviews and Meta-Analyses (PRISMA) protocol. PubMed and Web of Science (using the keywords “all databases”) were searched to identify eligible studies. The following keywords were used: “gestational diabetes mellitus” or “GDM” or “glucose” or “HOMA” or “insulin resistance “or “gestational diabetes” AND “pregnancy” or “maternal” or “pregnant” or “gestational” AND “probiotics” or “Lactobacillus” or “Bifidobacterium” or “probiotic”. Figure 1 presents the flow diagram of the review. 

### 2.4. Outcome Measures

The primary outcome was the influence of probiotic supplementation on the prevention and treatment of GDM (improvement in the glucose level).

## 3. Results

### 3.1. Role of Probiotics in Preventing Carbohydrate Disorders in Pregnant Women

#### 3.1.1. Research Evaluating the Effectiveness of Probiotics

Probiotics are live microorganisms that bring health benefits to the host when administered in appropriate amounts [36]. The role of probiotics in preventing carbohydrate disorders and GDM is not yet fully understood or proved; however, there are several studies that show promising results [37,38,39,40,41].

A study from New Zealand published in two papers (those of Barthow and Wickens et al.) [37,38] investigated the effect of supplementation with *Lactobacillus rhamnosus* HN001 from early pregnancy, without additional changes in the patient’s diet, on the incidence of GDM. It was a two-center, double-blind, randomized, parallel, placebo-controlled trial that eventually enrolled 423 pregnant women with a personal or partner history of atopic disease. Women were assigned to groups at 14–16 weeks of gestation. GDM diagnosis was a secondary outcome, and the women were assessed for GDM at 24–30 weeks of gestation. Lower relative indices of GDM (as defined by IADPSG criteria) were found for the *L. rhamnosus* group (RR 0.59; 95% CI 0.32, 1.08; *p* = 0.08). *L. rhamnosus* was associated with a lower percentage of GDM in women ≥35 years old (RR 0.31; 95% CI 0.12, 0.81; *p* = 0.009) and in women with a history of GDM (RR 0.00; 95% CI 0.00, 0.66; *p* = 0.004). However, these indicators were not significantly different from those of women without these issues. According to the definition provided by the New Zealand study, the frequency of GDM was significantly lower in the *L. rhamnosus* HN001 group—i.e., 2.1% (95% CI 0.6, 5.2)—than in the placebo group (*p* = 0.03); i.e., 6.5% (95% CI 3.5, 10.9).The results of the study indicate that the probiotic *L. rhamnosus* HN001 at a dose of 6 × 10^9^ CFUs per day could reduce the incidence of GDM by almost 40% according to the guidelines of IADPSG, especially among women over the age of 35 and those with prior GDM, or by up to 69% according to the New Zealand GDM diagnosis guidelines [37,38].

Another study that showed promising results was conducted on Mexican women (*n* =144). It assessed the influence of myo-inositol and probiotic supplementation (G-BALANCE^®^ containing 2 g myo-inositol + 5 × 10^8^ CFUs *Bifidobacterium lactis* and *L. rhamnosus*, administered twice a day between12–14and28 weeks of gestation) on the incidence of GDM in women with three or more risk factors for developing GDM (according to IADPSG criteria). The control group consisted of women undergoing regular prenatal care without supplementation with the above ingredients. The patients were matched according to age and body mass index (BMI). After the intervention, the frequency of GDM in the supplemented group was 29.2% and, in the control group, it was 47.9% (RR: 0.61; *p* = 0.03). The study concluded that supplementation with myo-inositol and probiotics starting at 12–14 weeks of pregnancy reduced the incidence of GDM in Mexican women [39]. Although the combination of myo-inositol and probiotics has been shown to be beneficial in preventing GDM, no clear conclusions can be drawn as to whether myo-inositol influences these results. The important fact is that a previous study confirmed the effectiveness of myo-inositol supplementation in reducing the incidence of GDM [42]. The exact mechanism of action of myo-inositol is unknown. However, the available hypotheses suggest the influence of myo-inositol on lipogenesis stimulation. It also increases glycogen synthesis and glucose uptake in peripheral tissues [42,43]. So far, few studies have been conducted in which the effects of supplementation based on the simultaneous supply of a probiotic and myo-inositol on the parameters of carbohydrate metabolism have been assessed, but their results indicate that administering both ingredients together does not have an additive, beneficial effect in GDM [43,44]. Further tests should be carried out using only probiotics to confirm their effectiveness.

A study by Luoto et al. [40] also suggests the efficacy of probiotics in preventing GDM. The impacts of dietary counseling combined with *L. rhamnosus* GG and *B. lactis* Bb12 supplementation on the outcomes of pregnancy and on infants (including the incidence of GDM in women) were assessed (*n* = 256). Two intervention groups received dietary advice; one took a placebo and the other a probiotic. The control sample consisted of women who were not given dietary advice. The results showed that the use of probiotics reduced the incidence of GDM. In total, GDM occurred in 13% of the women in the probiotic group, in 36% of the women in the placebo group, and in 34% of the women in the control group (*p* = 0.003). The risk of GDM was significantly reduced in the diet/probiotic group compared to that in the control group (OR = 0.27; 95% CI 0.11, 0.62; *p* = 0.002), while in the placebo group, the risk was not significantly different from that in the control group (OR = 1.08; 95% CI 0.55, 2.12; *p* = 0.823) [40]. In exactly the same group of patients, the effects of the above intervention on glucose metabolism and on serum insulin indices and insulin sensitivity were simultaneously assessed. The results indicated improved glucose metabolism and insulin sensitivity in the women tested. Among other things, the level of glucose was checked during pregnancy (mean values: 4.45 mmol/L in the probiotic diet, 4.60 mmol/L in the placebo diet, and 4.56 mmol/L in the control trial; *p* = 0.025) and at12 months after childbirth (mean values: 4.87 mmol/L in the probiotic diet, 5.01 mmol/L in the placebo diet, and 5.02 mmol/L in the control trial; *p* = 0.025).It turned out to be the lowest in the group that took probiotics. This group was also at the lowest risk of increased plasma glucose levels (*p* =0.013). Some of the subjects (45%) passed the glucose tolerance test. The probiotic group had the lowest incidence of pathological results. The group also showed improved insulin sensitivity compared to the control group (without dietary intervention) and the placebo group (the lowest insulin concentration (corrected means: 7.55, 9.32, and 9.27 m U/L; *p* = 0.032) and HOMA (corrected means: 1.49, 1.90, and 1.88; *p* = 0.028) and the highest quantitative insulin sensitivity check index (QUICKI; corrected means: 0.37, 0.35, and 0.35; *p* = 0.028)) in the last trimester of pregnancy. Diet has a huge impact on the microbiome and on certain diseases, such as GDM. Due to the lack of a control group with nutritional advice, the differences in the results cannot be clearly attributed to probiotics. The study seems to be promising, but it is not known exactly what influenced the results—nutritional recommendations, probiotics, or a combination of both [41].

Despite various studies confirming the efficacy of probiotics in preventing GDM [37,38,39,40,41], it still cannot be unanimously stated that this is actually an effective intervention. Some of the available studies agree with the hypothesis that probiotics effectively prevent carbohydrate metabolism disorders in pregnant women [45,46,47,48].

Callaway et al. [45] conducted a double-blind, randomized controlled trial comparing probiotics ((*L. rhamnosus* (LGG) and *B. animalis* subspecies *lactis* (Bb-12)) with a placebo in the prevention of GDM in overweight and obese pregnant women (*n* = 411) in Brisbane, Australia. Ultimately, GDM occurred in 12.3% of the patients (25 out of 204) in the placebo group and in 18.4% of the patients (38 out of 207) in the probiotic group (*p* = 0.10). In the OGTT conducted during the study, mean fasting blood glucose was higher in women randomized to the probiotics group compared to the placebo group (79.3 vs. 77.5 mg/dL; *p* = 0.049). A similar dependence was noticed in the measurements in the first and second hours of the test. In the study, the results of the probiotic group and the placebo group were similar, showing that probiotics did not prevent GDM in overweight and obese Australian women [45].

Another study (Pellonperä et al.) [46] that did not confirm the effectiveness of probiotic therapy in preventing GDM evaluated the effectiveness of interventions with fish oil and/or probiotics in preventing or treating GDM in overweight and obese women. This was a randomized, double-blind study with 439 women. The patients were divided into four parallel intervention groups: fish oil + placebo, probiotics + placebo, fish oil + probiotics, and placebo + placebo. Fish oil consisted of 1.9 g of docosahexaenoic acid (DHA) and 0.22 g of eicosapentaenoic acid (EPA), and probiotic supplements contained 10^10^ CFUs each of *L. rhamnosus* HN001 and *B. animalis* ssp. *lactis* 420. The percentage of women with GDM and the changes in glucose, insulin, or HOMA2-IR concentration did not differ between the intervention groups (*p* > 0.11 for each comparison). Thus, the study did not show any effectiveness of probiotics (neither fish oil nor a combination of both) in preventing GDM [46].

In a study by Lindsay et al. [47], the effect of a probiotic capsule on fasting glucose levels in obese pregnant women was compared to the effect of a placebo. The incidence of GDM was also one of the secondary outcomes. A probiotic capsule with *Lactobacillus salivarius* UCC118 at a dose of 10^9^ CFUs was used. In the study, no significant difference in the change in fasting glucose of women was found between the probiotic and placebo groups (−0.09 ± 0.27 vs. −0.07 ± 0.39 mmol/L; *p*= 0.391),nor any significant differences in the incidence of GDM and IGT. Ultimately, there were 6 cases of GDM (3 cases from each group) and 15 cases of IGT (7 cases in the probiotic group and 8 cases in the placebo group) (*p* = 0.561) [47].

#### 3.1.2. Summary

The importance of probiotics in preventing carbohydrate disorders in pregnant women is an area that requires more research. Although the results of some studies appear to be promising [37,38,39,40,41], there are also a number of studies that do not support the efficacy of using probiotics [45,46,47,48]. Thus, further research should be conducted to determine whether probiotic supplementation should be used extensively in early pregnancy to prevent GDM.

The available studies show divergent and inconclusive results. The heterogeneity of the results may have been related, inter alia, to the variety of probiotic preparations used, differences in the duration of exposure to probiotics, and the selection of patients based on various requirements. More homogeneous research is needed to determine an appropriate preparation containing specific probiotic strains [49]. There are many meta-analyses examining the effectiveness of probiotics, but they too do not provide conclusive results regarding the impact of probiotics in preventing GDM [48,50,51,52,53,54]. Table 2 summarizes the studies discussed in this section.

### 3.2. Role of Probiotics in Treating Carbohydrate Disorders in Pregnant Women

Currently, several studies are available on the use of probiotics in preventing GDM [37,38,39,40,41,45,46,47], but there are few studies on the efficacy of probiotics in treating carbohydrate disorders in pregnant women. However, we managed to locate five such studies [56,57,58,59,60]. Table 3 summarizes the studies discussed in this section.

The aim of one RCT conducted in Iran was to assess the effects of 8week multistrain probiotic supplementation (*L. acidophilus* LA-5, *B.* BB-12, *S. thermophilus* STY-31, and *L.delbrueckii bulgaricus* LBY-27) on indicators of glucose metabolism and changes in body weight in women with newly diagnosed GDM (*n* = 64) [56]. According to the analysis results, fasting blood glucose covariance in the intervention group and in the placebo group decreased significantly (from 103.7 to 88.4 mg/dlin the intervention group and from 100.9 to 93.6 mg/dl in the control group). However, the decrease in the probiotic group was significantly higher than in the placebo group (*p* < 0.05). The insulin resistance index in the probiotic group decreased significantly during the study period (by 6.74%). The insulin sensitivity index increased in both groups. These changes in the probiotic group (5.76%) were statistically significant. The researchers concluded that probiotic supplementation effectively improves glucose metabolism indices in patients with GDM. The results of this study are promising; however, the study should be conducted on a larger number of people not limited to Iranian women [56].

Another RCT, also conducted in Iran, investigated the effect of a synbiotic preparation (three viable and freeze-dried strains, *L. acidophilus*, *L. casei*, and *B. bifidum*, at a dose of 2 × 10^9^ CFUs/g each plus 800 mg of inulin) on markers of insulin metabolism and lipid profiles in GDM women (*n* = 70). After 6 weeks of intervention, compared to placebo, supplementation with the symbiotic preparation led to a significant decrease in serum insulin levels (−1.5 vs. 4.8 µIU/mL; *p* = 0.005), the HOMA-IR index (−0.4 vs. +1.1; *p* = 0.003), and HOMA-B (−5.1 vs. +18.9; *p* = 0.008) and a significant increase in QUICKI (+ 0.01 vs. −0.007; *p* = 0.02). The results suggest that symbiotic supplementation has favorable therapeutic potential for GDM patients, who often also exhibit with insulin resistance. More studies are needed on other patients, with larger groups, and over a longer time to determine the safety of this type of intervention. There is also a question of whether a probiotic would be as effective as a combination of a probiotic and a prebiotic [57].

One more RCT conducted in Iran examined the effects of probiotic supplementation on the genetic and metabolic profiles of patients with GDM. This study was conducted on a small group of 48 patients with GDM. The women were randomly divided into two equal groups that took either probiotic capsules containing *L. acidophilus, L. casei, B. bifidum,* and *L. fermentum* (2 × 10^9^ CFUs/g each) or a placebo for 6 weeks. The results showed that supplementation with probiotics significantly decreased the level of fasting glucose (−3.43 mg/dl; 95% CI −6.48, −0.38; *p* = 0.02), insulin concentration (−2.29 μIU/mL; 95% CI −3.60, −0.99; *p* = 0.001), and insulin resistance (−0.67; 95% CI −1.05, −0.29; *p* = 0.001) and significantly increased insulin sensitivity compared to the placebo (0.009; 95% CI, 0.004, 0.01; *p* = 0.001). The results indicate that supplementation with probiotics for 6 weeks in patients with GDM had a beneficial effect on glycemic control and insulin action. The study also showed a beneficial effect of supplementation on inflammatory markers, oxidative stress, and the expression of insulin-related genes; all of these elements may be involved in GDM development. This was thus another study that confirmed the effectiveness of probiotics in treating carbohydrate metabolism disorders in pregnant women [59].

In one study, the effects of 8weeks of intake of a probiotic mixture (VSL#3) on the glycemic status in women with GDM (*n* = 82) was investigated. VSL#3 is a freeze-dried pharmaceutical probiotic preparation containing 112.5 × 10^9^ CFUs/capsule of eight strains of bacteria (*S. thermophilus*, *B. breve*, *B. longum*, *B. infantis*, *L. acidophilus*, *L. plantarum*, *L. paracasei*, and *L. delbrueckii* subsp. *Bulgaricus*). A comparison between the two groups showed no significant differences in several glycemic parameters, such as fasting blood glucose and HbA1c (*p* > 0.05), but significant differences in insulin levels and HOMA-IR (16.6 ± 5.9; 3.7 ± 1.5; *p* = 0.03). The study suggests that supplementation with various subspecies of *Lactobacillus*, *Bifidobacterium*, and *Streptococcus* may benefit glycemic status. In the study, the use of the supplement did not significantly influence fasting glucoseorHbA1c but it prevented an increase in serum insulin concentration and resulted in an improvement in insulin resistance [60].

A study conducted in Thailand examined the effects of 4weeks of probiotic therapy on insulin resistance in women with newly diagnosed GDM (*n* = 60). One group of women received the Infloran^®^ probiotic, each capsule containing 10^9^ CFUs of *L. acidophilus* and 10^9^ CFUs of *B. bifidum*. Changes in metabolic parameters after randomization showed a significant improvement in glucose metabolism in the probiotic group compared to the placebo group, including improvements in fasting glucose (0.68 ± 5.88 vs. 4.620 ± 7.78 mg/dL, MD −3.94 mg/dL; 95% CI −7.62, −0.27; *p* = 0.034), fasting plasma insulin (1.11 ± 1.71 vs. 3.77 ± 1.70 mIU/L, MD −2.67 mIU/L; 95% CI −3.57, −1.76; *p =* 0.001), and the HOMA-IR index (0.25 ± 0.37 vs. 0.89 ± 0.46, MD −0.63; 95% CI −0.86, −0.41; *p* = 0.001). In addition, probiotic supplementation was found to be well-tolerated and safe in the participants. These results confirmed the beneficial effect of probiotics on glucose metabolism in women with GDM in Thailand [58].

All of the above studies support the beneficial effects of probiotics in treating carbohydrate metabolism disorders in pregnant women, but most of the studies were carried out in Iran, and more studies are needed to cover wider groups of patients. A few of the studies included small groups from specific countries, and the research results are promising and suggest that probiotics may effectively treat GDM [56,57,58,59,60]. A meta-analysis from November 2018 showed that probiotic supplementation significantly reduced fasting glucose, insulin resistance, and insulin concentration in pregnant women, not only in those not diagnosed with GDM but also in women with GDM already diagnosed, proving the potential role of probiotics in both the prevention and treatment of carbohydrate metabolism disorders [51]. Two meta-analyses carried out consecutively in 2020 and 2022 showed that probiotics improve carbohydrate metabolism and insulin activity indicators, such as HOMA-IR and QUICKI, and can reduce the risk of insulin resistance in patients with GDM [53,61]. Among the existing studies, there was noticeable heterogeneity among the interventions used, making it difficult to compare the studies.

Nevertheless, the results do indicate that probiotic supplementation seems to improve the functioning of glucose metabolism and reduce the risk of GDM in women. However, uncertainty remains due to the heterogeneity of the existing studies. More homogeneous studies are needed to declare the effectiveness of probiotics confidently [50].

### 3.3. The Mechanism of Action of Probiotics

During pregnancy, women’s risk of secreting pro-inflammatory cytokines, such as leptin, resist in, IL-6, IL-10, TNF-α, interferon-gamma (IFN-γ), and C-reactive protein (CRP), becomes significantly higher and the elevated concentrations of these cytokines, combined with elevated placental lactogen, progesterone, and estrogen, can increase insulin resistance and glucose intolerance considerably [62]. Furthermore, the inflammatory process is incidental to oxidative stress conditions caused by hyperglycemia, which arean effect of the overproduction of reactive oxygen species (RFT) and/or a shortage of antioxidant defense systems [63]. Not only does it cause many pathophysiological complications, it also shows a close relationship with insulin resistance, leading to reduced glucose uptake in peripheral tissues and increased glucose production in the liver [64].

Research shows that the composition of the GM in pregnant women changes in a significant way. In the last trimester specifically, there is a sudden decrease in bacteria essential for the regulation of metabolism and an increase in Proteobacteria and Actinomycetes, which lead to the development of the inflammatory condition mentioned above. What is more, the amounts of accumulated fat and stored nutrition depend on the condition and composition of the microflora, and the related disorders often lead to the formation of easily digestible monosaccharides and lipoprotein lipase activation as a result of hydrolysis of undigested polysaccharides. The consequence is excessive storage of substances of hepatic origin (triglycerides) [65]. Therefore, microflora homeostasis dysfunctions of all types can be the direct cause of diabetes mellitus and anomalies in the level and composition of SCFAs, resulting in energy metabolism disorders, eating disorder, or disorders related to blood glucose homeostasis [63]. Results from multiple studies verified that consumption of probiotics by pregnant women with GDM can help significantly control their glycemia and glucose metabolism (as well as leading to a crucial decline in HOMA-IR). It can also reduce the levels of VDL cholesterol, triglycerides, and even inflammatory markers. The mechanism has not been explained yet, so it requires further research [62]. Probiotics produce benefits mainly by restoring correct microflora, normalizing increased intestinal permeability, and regulating the secretion of pro-inflammatory mediators [65]. Insulin resistance biomarkers can be influenced by anti-inflammatory probiotic features and increased production of bacteriocins and SCFAs (e.g., butyrate, propane, and acetate), which work as chemical mediators by passing information from the intestinal lumen on to the rest of the body and regulating energetic metabolism and the expansiveness of fat tissue [62,64]. As an example, butyrate, which takes part in mucus secretion and in supporting the regulating features of T lymphocytes, helps strengthen intestinal mucosa’s protective barrier and temper inflammatory reactions [63]. Moreover SCFAs produced by probiotics can decrease hs-CRP levels in serum by blocking the enzymatic synthesis of CRP in the liver, which is activated in response to factors such as IL-6 [64]. The antioxidant features of probiotics presumably result from reduced lipid peroxidation and the resulting increase in antioxidant levels or engagement with enzymes, such as glutathione s-transferase, glutathione peroxidase, glutathione reductase, superoxide dismutase, and catalase [63]. Probiotics may defend against oxidative stress by secreting peptides, restoring normal intestinal flora, and removing oxidizing compounds or simply preventing their formation in the bowel [64].

### 3.4. Clinical Implications

As mentioned above, probiotics may contribute to glycemic stabilization. Both probiotic therapy and the supply of prebiotics can be important elements in preventing and treating carbohydrate disorders in pregnancy.

Research has shown the potential of these substances; however, for clinical practices, it is crucial that we find answers to the questions that have not yet been resolved. There is still much research required before we can routinely recommend probiotics in the treatment of women at risk of GDM.

The secondary results of the study described by Barthow and Wickens et al. [37,38] regarding the effect of probiotic supplementation on the occurrence of GDM are promising and indicate the effectiveness of *L. rhamnosus* HN001 in the prevention of GDM. However, people who had had an atopic disease in the past were recruited for the study and it is not known whether the obtained results would be the same if a healthy population was taken into account, without the risk of atopic disease or allergy. The study by Reyes-Muñoz et al. [39] was conducted on a smaller group of patients than the study described above and used a formulation that combined a probiotic with myo-inositol. Although the results indicated the efficacy of the preparation, a similar study using the probiotic alone is needed to determine whether it would also reduce the risk of GDM on its own. The studies by Luoto and Laitinen [40,41] also showed a beneficial effect of probiotics on the occurrence of GDM and the regulation of carbohydrate metabolism. However, in this study, an additional intervention was dietary counseling in patients and this intervention contributed to the results, and although the most favorable values were obtained in the group with probiotics and dietary counseling, it is clear that this was not solely due to the probiotics. Here, efficacy cannot be attributed solely to probiotics, as adequate nutrition has a huge impact on microbiota. The study by Callaway et al. [45] indicated a lack of efficacy for probiotics in the prevention of GDM; however, a study on a larger population other than Australians is needed to confirm these results in women around the world. Similarly, Pellonperä et al. [46] and Lindsay et al. [47] did not confirm the efficacy of probiotics in the prevention of GDM in their study; however, the intervention was conducted in obese and overweight women, which may have influenced the results, and further studies on women with adequate body weight are needed to compare the effects of probiotics in these two populations.

The heterogeneity of the results may be related, inter alia, to the variety of probiotic preparations used, the different durations of exposure to probiotics, and the selection of patients based on the various requirements. For example, the New Zealand study used *L. rhamnosus* HN001 from early pregnancy and the Luoto study used *L. rhamnosus* GG in combination with *B. Lactis* BB12 [37,38,40].

As regards the effects of probiotics on the treatment of carbohydrate disorders in pregnant women, the results are more conclusive, but more research is still needed to determine the appropriate strains of bacteria and their dosage. Most of the available studies were conducted in Iran, so further studies are needed on a wider population.

Dolatkhah et al. [56] showed that supplementation with a multi-strain probiotic reduced fasting blood glucoseand HOMA-IR and increased QUICKI. Another Iranian study (Ahmadi et al.) [57] demonstrated beneficial effects of synbiotic supplementation on markers of insulin metabolism (serum insulin levels, HOMA-IR, HOMA-B, and QUICKI). However, the effects of the probiotic combined with the prebiotic could have been different than if the probiotic by itself was used, so the obtained results cannot be attributed solely to probiotics. The results of the study by Babadi et al. [59] also indicated the efficacy of probiotic therapy in regulating glycaemia and insulin action, but this study was conducted on a group with only 48 patients and is difficult to translate to a larger population. Jafarnejad et al. [60], in their study, showed an effect from probiotics in increasing serum insulin levels and improving insulin resistance, but their results did not confirm effects on parameters such as FBG and HbA1c. As mentioned earlier, further research is needed on other populations outside Iran. One study conducted outside Iran was conducted in Thailand (Kijmanawat et al.) [58]. This study also confirmed beneficial effects on glucose metabolism parameters (fasting glucose, fasting plasma insulin, HOMA-IR). The above studies also used various preparations with different strains of bacteria (Jafarnejad—different strains:VSL#3–8; Kijmanawat—Infloran^®^; Babadi—a mixture of strains: *L. acidophilus*, *L. casei*, *B. bifidum*, and *L. fermentum;* Ahmadi– a synbiotic; Dolatkhah—a mixture of strains: *L. acidophilus* LA-5, *B.* BB-12, *S. thermophilus* STY-31, and *L. delbrueckii bulgaricus* LBY-27) [56,57,58,59,60]), which does not allow for a consistent appraisal of which strains are the most effective and have an impact on carbohydrate management.

#### 3.4.1. Dose and Strain of Bacteria in Terms of the Effectiveness of Probiotic Therapy in Women with GDM

Probiotics can reduce some of the negative metabolic side effects experienced by pregnant women. Supplementation can improve glycemic control for women who are in their third trimester. It can also lower the risk of GDM, which is suspected to be mainly achieved by lowering insulin resistance [55]. Dosage is an important factor influencing the effectiveness of probiotic supplementation in the metabolic health of pregnant women. A dosage higher than 10^7^ CFUs/day showed a positive impact on the metabolic health of a pregnant woman [64]. As discussed above, the probiotic *L. rhamnosus* HN001 administered in dosages of 6 × 10^9^ CFU/day can lower the prevalence of GDM. A similar supplementation with myo-inositol and probiotics *L. rhamnosus GG* and *B. lactis* BB12 from the 12th to the 14th weeks of pregnancy can improve glucose metabolism and insulin sensitivity. Supplementation with *L. acidophilus, L. casei, B. bifidum,* and *L. fermentum* (2 × 10^9^ CFUs each) can effectively lower fasting blood glucose levels. Administration of 10^9^ CFUs of *L. acidophilus* and 10^9^ CFUs of *B. Bifidum* can improve glucose metabolism. The above data are supported by research results. However, we still lack absolute certainty that this is indeed an effective intervention. Further research is required to determine the optimal probiotic dose for pregnant women. There are also studies that exclude the influence of probiotics on carbohydrate metabolism.

#### 3.4.2. The Safety of Probiotic Use

Probiotics and prebiotic products are safe to use during pregnancy and lactation. Among other things, thanks to the Generally Recognized As Safe(GRAS) status, consumers can be informed that the microorganisms used in probiotics are not hazardous to health. Similarly, other governments and associations around the world (e.g., EFSA) have regulatory protocols, which vary from country to country. Considering their low and chronic toxicity, probiotics are considered safe. Of course, preparations used by pregnant women must be of high quality, contain tested strains, and be free from contamination [66]. Some studies have shown that intake of *L. rhamnosus* and *L. reuteri* was associated with an increased risk of vaginal discharge and changes in stool consistency. Nevertheless, the adverse effects of probiotic supplementation during pregnancy are not life-threatening and do not pose serious health problems for mother or child [67].

#### 3.4.3. Additional Benefits of Probiotic Therapy for Pregnant Women

*Lactobacillus* and *Bifidobacterium* have beneficial effects on the metabolic health of pregnant women. Lactic acid bacteria exhibit antioxidant properties, reducing the oxidative stress associated with hyperglycemia, which often occurs during pregnancy. In addition, probiotics can reduce the levels of interleukin-6 (IL-6), tumor necrosis factor (TNF-α), and high-sensitivity C-reactive protein (hs-CRP), thereby reducing inflammation in the body. Pregnant women can also balance the properties of abnormal native intestinal microflora using probiotics. Research on other strains and their effects on the health of pregnant women is limited, so more research is needed in this direction [64]. Another probiotic of importance to pregnant women is *B. lactis*, which can improve the intestinal transit time, frequency, and consistency of stools in the case of constipation, which pregnant women often face [68]. Intimate infections are a common problem for pregnant women. In such cases, probiotics such as *L. fermentum* 57A, *L. plantarum* 57B, and *L. gasseri* 57C are recommended. They improve values such as the vaginal pH and Nugent Score. They also positively affect the total number of vaginal lactobacilli [69].

Consumption of a probiotic by mothers can also benefit their babies. Supplementation in the last month of pregnancy and the administration of a probiotic to an infant for the next 6 months of life reduce the risk of allergies in a child. Administration of the extensively hydrolyzed casein preparation with the addition of *L. rhamnosus* GG results in accelerated tolerance development in children allergic to cow’s milk, as the strain affects the structure of the infant’s intestinal microflora [70]. Despite the fact that GDM is a common cause of macrosomia in newborns, more research is needed on the impact of probiotics in reducing this risk [71]. So far, no anthropometric changes have been demonstrated in newborns whose mothers consumed probiotics during pregnancy. It is important to conduct studies that consider the total birth weight, birth weight percentile, macrosomia, head circumference, and length of the newborn [64]. The study also noted that the use of probiotics did not reduce the risk of miscarriage or stillbirth in pregnant women [71]. More research is needed on the effects of probiotics on fasting glucose and low-density lipoprotein levels [55]. No significant correlation was found between probiotic intake by women with GDM and reductions in total cholesterol, low-density lipoprotein, high-density lipoprotein, and triglycerides. It is also worth conducting a study examining the influence of probiotics on metabolic changes in obese pregnant women and pregnant women with normal body weight [64].

#### 3.4.4. Summary

In sum, supplementation with probiotics may positively affect glycemic control but it does not affect the level of blood lipids in pregnant women with GDM. Further studies should be carried out to establish and compare the efficacy of different probiotic strains and different doses of CFUs. These interventions should determine the optimal dose and the specific strain recommended for supplementation in cases of healthy pregnant women and pregnant women with GDM [64].

## 4. Conclusions

On the basis of the research presented in this article, it can be concluded that supplementation with probiotics positively affects glycemic control (lowering FBG, serum levels, and HOMA-IR) and may lower the frequency of GDM. Although the results of some studies on GDM prevention (Wickens et al., Reyes-Muñoz et al., and Luoto et al.) and treatment (Ahmadi et al., Kijmanawat et al., Babadi et al.,Jafarnejad et al., and Dolatkhah et al.) appear to be promising, further studies must be conducted to determine the mode of action of probiotics in pregnant women with GDM. The available studies provide divergent and inconclusive results. The heterogeneity of the results may be related, inter alia, to the variety of probiotic preparations used, different exposure times to probiotics, and the selection of patients based on different requirements. Therefore, future studies should aim to standardize the study group, strain, and dose of probiotics so that, on the basis of reliable data and results, probiotics can be safely implemented in the prevention and treatment of GDM in pregnant women.

## Figures and Tables

**Figure 1 nutrients-14-04303-f001:**
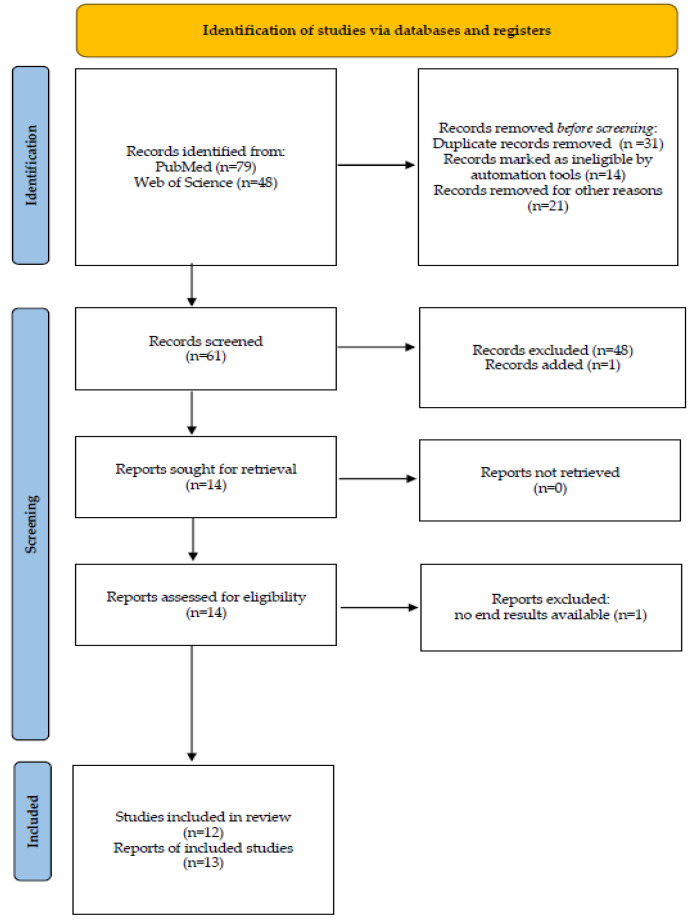
Flow chart of the study.

**Table 1 nutrients-14-04303-t001:** The differences in the GMs of gestational diabetes mellitus (GDM) and normoglycemic pregnant women.

References	Population	Differences in the GM
Kuang et al. [27]	GDM women (*n* = 43) vs. healthy pregnant women (*n* = 81)	In GDM: ↑*Parabacteroides distasonis* and *Klebsiella variicola* ↓*Methanobrevibactersmithii*, *Alistipes*, *Bifidobacterium*, and *Eubacterium*
Crusell et al. [7]	GDM (*n* = 50) vs. healthy pregnant women (*n* = 157) in the third trimester	In GDM: ↑Actinobacteria, *Collinsella*, *Rothia*, *Actinomyces,* and *Desulfovibrio* ↑*Blautia* and *Ruminococcus* ↓*Acetivibrio*, *Intestinimonas*, *Erysipelotrichaceaeincertaesedis*, *Isobaculum*, *Butyricicoccus*, *Clostridium IV*, *Clostridium XVIII*, *Oscillibacter*, *Ruminococcus*, *Bacteroides*, *Veillonella*, and *SuterellaFaecalibacterium*
Cortez et al. [28]	GDM (*n* = 26) vs. non-GDM (*n* = 42) women in the third trimester of gestation	In GDM: ↓Diversity ↑*Lachnospiraceae, Phascolarctobacterium*, and *Christensenellaceae* ↑*Ruminococcus, Eubacterium*, and *Prevotella* ↓*Bacteroides*, *Parabacteroides*, *Roseburia, Dialister*, and *Akkermansia*
Ye et al. [29]	Pregnant women (*n* = 52) in the third trimester: non-GDM women and women with GDM but with successfully controlled blood glucose(GDM1)vs. GDM women with uncontrolled blood glucose(GDM2)	In GDM: ↑*Blautia* and *Eubacteriumhalliigroup* ↓*Faecalibacterium, Subdoligranulum*, *Phascolarctobacterium*, and *Roseburia* No significant differences in GM composition and a difference in the relative abundance between the GDM1 and N groups
Zheng et al. [30]	GDM women (*n* = 31) vs. healthy pregnant women (*n* = 103)	In GDM: ↓*Coprococcus* and *Streptococcus* Lower number of dynamic changes in gut microbiota in the first half of pregnancy predisposing to the development of GDM
Mokkala et al. [31]	Overweight/obese women with GDM (*n* = 270)	In GDM: ↑*Ruminococcusobeum*

**Table 2 nutrients-14-04303-t002:** Characteristics of randomized controlled trials included in Section 3.1. [37,38,39,40,41,45,46,47,55].

Author/Year	Participants	Study Design	Intervention	Effect of Dietary Probiotic Supplement on Outcomes
Laitinen et al. (2008) [41]	Pregnant women; no chronic diseases apart from allergic diseases; less than 17 weeks of gestation (gw) (*n* = 256)	Parallel RCT	Random assignment to a control ora dietary intervention group. The intervention group received intensive dietary counseling provided by a nutritionist, and the women were further randomized, double-blind, to receive probiotics (*L. rhamnosus* GG and *B. lactis* Bb12at a dose of 10^10^ CFUs/day each; diet/probiotics) or a placebo (diet/placebo)	↓Plasma glucose levels ↓Risk of elevated plasma glucose levels ↓Frequency of pathological results in the glucose tolerance test ↓Insulin concentration ↓HOMA-IR ↑QUICKI
Luoto et al. (2010) [40]	Pregnant women; no chronic diseases apart from allergic diseases; less than 17 gw (*n* = 256)	Parallel RCT	Random assignment to a control ora dietary intervention group. The intervention group received intensive dietary counseling provided by a nutritionist, and the women were further randomized, double-blind, to receive probiotics (*L. rhamnosus* GG and *B. lactis* Bb12 at a dose of 10^10^ CFUs/day each; diet/probiotics) or a placebo (diet/placebo)	↓Prevalence of GDM
Lindsay et al. (2015) [47]	Women with GDM (*n* = 149)	Parallel RCT	Random assignment to 6week probiotic or placebo capsules. Each capsule contained *L. salivarius* UCC118 (1 × 10^9^ CFUs/g)	⟷ Fasting blood glucose ⟷ HOMA-IR ⟷ C-peptide No significant effect on the incidence of GDM
Wickens et al. (2017) [38]	Pregnant women with a personal or partner history of atopic disease (*n* = 423)	Parallel RCT	Random assignment to probiotic or placebo capsules between14–16 and24–30 weeks of gestation. Each probiotic capsule, administered daily, contained *L. rhamnosus* HN001 at a dose of 6 × 10^9^ CFUs	↓Prevalence of GDM
Callaway et al. (2019) [45]	Overweight and obese pregnant women; less than 20 gw (*n* = 411)	Parallel RCT	Random assignment to probiotic or placebo capsules. Each probiotic capsule contained *L. rhamnosus* (LGG) and *B. animalis* subspecies *lactis* (BB-12) at a dose of >1 × 10^9^ CFUs and was administered daily from enrolment until birth	No significant effect on the incidence of GDM
Pellonperä O et al. (2019) [46]	Overweight and obese pregnant women; less than 18 gw; absence of chronic diseases (asthma and allergies were allowed) (*n* = 439)	Parallel RCT	Random assignment to one of four parallel groups: fish oil + placebo (i.e., placebo for probiotics), probiotics + placebo (i.e., placebo for fish oil), fish oil + probiotics, and placebo + placebo (i.e., placebo for probiotics and placebo for fish oil). The fish oil capsules contained a total of 2.4 g of *n*-3 fatty acids, of which 79% (1.9 g) were docosahexaenoic acid (22:6; *n*-3) (DHA) and 9.4% (0.22 g) eicosapentaenoic acid (EPA). The probiotic capsules contained *L. rhamnosus* HN001 and *B. animalis* ssp. *lactis* 420, 10^10^ CFUs per capsule. Supplements were provided from the first study visit throughout the pregnancy until 6 months postpartum	No significant effect on the incidence of GDM
Reyes-Muñoz et al. (2021) [39]	Mexican women with three or more risk factors for developing GDM (*n* = 144)	Parallel RCT	Random assignment to probiotic with myo-inositol or placebo capsules between12–14 and28 weeks of gestation. Each dose contained myo-inositol 2 g + *B. lactis* and *L. rhamnosus* 5 × 10^8^ CFUs and was administered twice a day	↓Prevalence of GDM

Abbreviations: BMI, body–mass index; CFUs, colony-forming units; DHA, docosahexaenoic acid; GDM, gestational diabetes mellitus; gw, weeks of gestation; HOMA-IR, homeostasis model assessment of insulin resistance; RCT, randomized controlled trial; QUICKI, quantitative insulin sensitivity check index

**Table 3 nutrients-14-04303-t003:** Characteristics of randomized controlled trials included in Section 3.2 [55,56,57,58,59,60].

Author/Year	Participants	Study Design	Intervention	Effect of Dietary Probiotic Supplement on Outcomes
Dolatkhah et al. (2015) [56]	Pregnant adult Iranian women with GDM (*n* = 64)	Parallel RCT	Random assignment to 8week probiotic or placebo capsules. Each probiotic capsule contained four bacterial strains (4 biocaps > 4 × 10^9^ CFUs)—i.e., *L.acidophilus* LA5, B. BB-12, *S.thermophilus* STY-31, and *L. delbrueckii bulgaricus* LBY-27—in a standard freeze-dried culture	↓Fasting blood glucose ↓HOMA-IR
Ahmadi et al. (2016) [57]	Pregnant adult Iranian women with GDM without previous diagnoses of diabetes at 24–28 weeks of gestation (*n* = 70)	Parallel RCT	Random assignment to 6week synbiotic or placebo capsules. The synbiotic capsules contained *L.acidophilus*, *L. casei*, and *B.bifidum* (2×10^9^ CFUs/g each) plus 0.8 g of inulin	↓Serum insulin levels ↓HOMA-IR ↑QUICKI
Jafarnejad et al. (2016) [60]	Pregnant adult Iranian women with GDM (*n* = 82)	Parallel RCT	Random assignment to 8week probiotic or placebo capsules. Each probiotic capsule contained VSL#3 (*S. thermophilus*, *B. breve*, *B. longum*, *B. infantis*, *L. acidophilus*, *L. plantarum*, *L.paracasei*, and *L.delbrueckii* subsp. *Bulgaricus*; 15 × 10^9^ CFUs/g)	↓Fasting plasma glucose ↓HOMA-IR
Kijmanawat et al. (2019) [58]	Pregnant Thai women with GDM (*n* = 60)	Parallel RCT	Random assignment to 4week probiotic or placebo capsules. Infloran^®^ probiotic was used, each capsule containing 1000 million CFUs of *L. acidophilus* and 1000 million CFUs of *B. bifidum*	↓Fasting plasma glucose ↓Serum insulin levels ↓HOMA-IR
Babadi et al. (2019) [59]	Pregnant adult Iranian women with GDM who were not on oral hypoglycemic agents (*n* = 48)	Parallel RCT	Random assignment to 6week probiotic or placebo capsules. Each probiotic capsule contained *L.acidophilus*, *L. casei*, *B.bifidum*, and *L. fermentum* (2 × 10^9^ CFUs/g each)	↓Fasting plasma glucose ↓Serum insulin levels ↓HOMA-IR ↑QUICKI

Abbreviations: BMI, body–mass index; CFUs, colony-forming units; GDM, gestational diabetes mellitus; HOMA-IR, homeostasis model assessment of insulin resistance; RCT, randomized controlled trial; QUICKI, quantitative insulin sensitivity check index.

## Data Availability

Not applicable.

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
