# Peer review of "Probiotics in the Prevention and Treatment of Gestational Diabetes Mellitus (GDM): A Review"

_nutrients, 2022, doi:10.3390/nu14204303_

Round 1

Reviewer 1 Report

This review focused on the use of probiotics for the prevention and treatment of gestational diabetes. The review provides a detailed description of gestational diabetes, the incidence, the changes in the host immune system and the pregnant mothers microbiome during each trimester of pregnancy. The authors review studies where probiotics were used either to prevent gestational diabetes or treat geastational diabetes. The studies are very well described and the authors provide a balanced overview being cognisant of the different geographical locations of the studies, the study groups and the probiotic interventions used. It was not immediately clear how the authors came to the final 12 studies that were incorporated in the review and the exclusion criteria that were used. Indeed many more studies were excluded than included in the review. Finally the authors propose mechanisms by which probiotics may modulate the microbiome to be efficacious in gestational diabetes. The authors highlight that much more research is required and it is not yet justified to use probiotics to prevent or treat gestational diabetes. Overall this is a good educational and thorough review of the field of gestational diabetes and of the selected intervention studies. I would suggest that the introduction be reviewed to improve  english and clarity.

Author Response

Dear Reviewer,

Thank you for the review of the manuscript. We are grateful for the comment to our paper. We have carefully reviewed your comment and as You suggest, our manuscript has been rewritten by MDPI English Editing Service (certificate in attached file). Major changes to the text are highlighted “yellow” in the revised manuscript. We hope the revised version is now suitable for publication and look forward to hearing from you in due course.

Sincerely,

Małgorzata Moszak

Reviewer 2 Report

Gestational diabetes is a problem derived from pregnancy that can trigger serious problems in mother and child. Therefore, the work that can bring together the knowledge generated is of value. There are already fairly recent state-of-the-art reviews (Homayouni et al. 2020; Davidson et al., 2021; Taylor et al., 2017; Zhou et al., 2021; Owens , 2021). Still, an extensive state-of-the-art review is always welcome if it shares a high quality of data and analysis. In this review, we analyze not only diabetes but also studies related to other carbohydrate-related parameters. However, there are points that need to be strongly improved.

Major changes

The English is not accurate in writing, and there are a lot of grammatical mistakes. The paper should be corrected by a professional native English proofreader.

In 3.1.1. it is included “The role of probiotics in preventing carbohydrate disorders and GDM is still not fully understood and not justified;” why are they still not justified?” can the authors include just some words with the main reason?

3.1. Probiotics in the prevention of carbohydrate disorders in pregnant women

In the first study included:

-           (the cite is not added at the beginning of the paragraph, please include)

-          - GDM was not the primary outcome and not related initially with this disease.

-          Also women carried a primary condition previously related with gut-skin axis. So that, the robustness to be highlighted is weak.

The second study (Mexico) was carried out with a product combining myo-.inositol and probiotics.

-          Can the authors cite the strains?

-          Can the authors discuss about the extrapolation of priobiotics’ functionality and/or how myo-inositol is influencing the results, based on previous data?

-          Also, the authors indicated that larger sample size is necessary, but the number of women included are not cited in the review.

The third study (Luoto)

-          Could the authors extrapolated the results when the control group did not have dietary advice? Diet impacts extremely on microbiome, and of course in specific diseases, such as GGDM. Due to the lack of a control group with the dietary advice, difference can not be ascribed to probiotics. This needs to be included in the discussion of this paper. Based on that, the statement “Among other things research results may indicate a significant effect of probiotics on carbohydrate metabolism in the body, and thus the development and prevention of GDM [41” is not true in this case, please remove.

Callaway et al:

-          Please included the strains, not only species (if available)

The fifth study:

-          Please include firstly reference or authors.

-          Was it a cross-desing? Parallel? Please inform

-           

Could the sentence “Despite the fact that there are studies confirming the effectiveness of probiotics in preventing GDM (3–6), it still cannot be unanimously stated that this is actually an effective intervention. Some of the available studies agree with the hypothesis that probiotics effectively prevent carbohydrate metabolism disorders in pregnant women [42–44].” Be reallocated in other position that does not interfere with the flow of the paper?

Although the authors cite and describe different clinical trials, there is not any  comparison nor discussion at all. The authors should discuss at least finally some of the results found, not only a (very brief) summary of the data included.

3.2 Probiotics in the treatment of carbohydrate disorders in pregnant women.

Again, please indicate the Table first of all, to guide the readers.

Iranian study:

-          The results of this study are promising; however, the study should be conducted on a larger number of people and not be limited to Iranian women [53]. In any case, the results should not be limited to an specific country, so it is not an specific conclusion or discussion for this study. If this is the conclusion of the authors, please cite in this way.

-          “More studies are needed in other patients, in larger groups, and over a longer time to determine the safety of this type of intervention” why safety is included? Initially, these initial CTs are designed to ensure the safety of the products. was this not achieved in the first study due to design? Or due to some issues in the project?

-           The second one, must be considered as a pilot, isn’t it?

The #VSL3 :

-          HOMA-IR differences are not included in tehxt.

In all these studies, there were in any microbiome analysis? If it was, please include.

As not gut microbiome data linking probiotics with gut microbiome and improvement, the sentence “Nevertheless, the results declare that by regulating the GM, probiotic supplementation seems to improve” is not really drafted demonstrated

“Thanks to the GRAS status (Generally Recognized As Safe), consumers can be confident that the microorganisms used in probiotics are not pathogenic” GRAS status is assigned to specific strains, and takes into account not only the strain but also the product information, so this is not as general as cited. Moreover, other governments  or associations in the world (EFSA is an example) have other regulatory protocols. So that, GRAS status in US (of specific strains) does not extrapolate to the rest of probiotics.

MInor:

Examples of grammatical incorrections:

·         Abstract please change: “ In the pathogenesis of GDM genetics, environmental, and pregnancy-related factors (excessive fat storage, adipokine, and increased cytokine secretion) play an important role.” Or “Genetics, environmental, and pregnancy-related factors (excessive fat storage, adipokine, and increased cytokine secretion) play an important role in the pathogenesis of GDM”

·          Abstract please change: “A growing number of scientific data report the role of …” or A growing number of scientific data has reported the role of …”

·         Abstract please change: “ and a lower number of genera Bacteroides, Parabacteroides, Roseburia, Dialister, and Akkermansia”.

·          Abstract please change: “analyzed data”

·          Abstract please change: “probiotics positively affect glycemic…”

·         Introductin, please change:  “the blood volume increases”

·         1.1. “  activity have been observed [26]”

Introductin, please change:   

·          Pregnancy influences hyperventilation in the respiratory system [2], decreases functional residual capacity, and modulates inspiratory reserve volume.

·     GDM development has also been proved [27,28”

·      In 3.2.1. According to the analysis results please change to “According to the results”

·          

Part 1.1.  Please correct “in phyla Proteobacteria and Actinobacteria”. Phylum level is without italics. Please correct in all the manuscript

Why do the authors include 1.1. in Introduction, if there are no more divisions? Maybe initially it was 1.1. GDM disease and 1.2. GDM microbiota? In any case, please correct

Part 1.1.  The relationship between changes in the abundance of Blautia, Butyricicoccus, Clostridium, Coprococcus, Dorea, Faecalibacterium, Ruminococcus, and Lach (reduced numbers) and Collinsella and Rikenallaceae (increased numbers). What is “Lach” is it Lachnospira” please check and correct

Table 1. “higher abundance of Ruminococcus, Eubacterium, and Prevotella “ needs an arrow? In general, Table 1 has sentences that seem directly taken from the papers, the text is not standardized (in some cases appear the word genera or species, in some not, in some appear “and”, in some not…) please correct the Table to be accurately exposed. Also in Table 1, the characteristics of the population should be included in a more legible way, in different columns (it is hard to compare the different studies with all the information in the same column “n, age, etc” or at least with the same order in all cases).

Please correct “Importantly, scientific data show that the state of dysbiosis during GDM can be modified through dietary intervention”.

In 3.1.1. the name of the strains can’t be cited in italics. Please correct to “HN001” and subsequents.

As a minor, please try not to conclude a paper with the citation of a Table. The sentence “Table 2 summarizes the studies in this section.” Can be cited at the beginning of the section helping readers a better understanding of the results.

3.2. What is FBS? I think it is not introduced in the text before

Round 2

Reviewer 2 Report

The review has improved a lot, considering now Ok for its publication